# Assessment of Thermal Stresses in Asphalt Mixtures at Low Temperatures Using the Tensile Creep Test and the Bending Beam Creep Test

**Marek Pszczola ***, **Mariusz Jaczewski** and **Cezary Szydlowski**

Faculty of Civil and Environmental Engineering, Gdansk University of Technology, 80-233 Gdansk, Poland;
marjacze@pg.edu.pl (M.J.); cezszydl@pg.edu.pl (C.S.)
\* Correspondence: mpszczol@pg.edu.pl; Tel.: +48 58 347-27-82

**Abstract:** Thermal stresses are leading factors that influence low-temperature cracking behavior of asphalt pavements. During winter, when the temperature drops to significantly low values, tensile thermal stresses develop as a result of pavement contraction. Creep test methods can be suitable for the assessment of low-temperature properties of asphalt mixtures. To evaluate the influence of creep test methods on the obtained low-temperature properties of asphalt mixtures, three point bending and uniaxial tensile creep tests were applied and the master curves of stiffness modulus were analyzed. On the basis of creep test results, rheological parameters describing elastic and viscous properties of the asphalt mixtures were determined. Thermal stresses were calculated and compared to the tensile strength of the material to obtain the failure temperature of the analyzed asphalt mixtures. It was noted that lower strain values of creep curves were obtained for the Tensile Creep Test (TCT) than for the Bending Beam Creep Test (BBCT), especially at lower temperatures. Results of thermal stress calculations indicated that higher reliability was obtained for the viscoelastic Monismith method based on the TCT results than for the simple quasi-elastic solution of Hills and Brien. The highest agreement with the TSRST results was also obtained for the Monismith method based on the TCT results. No clear relationships were noted between the predicted failure temperature and different methods of thermal stress calculations.

**Keywords:** asphalt mixture; low-temperature cracking; Tensile Creep Test (TCT); Bending Beam Creep Test (BBCT); tensile strength; thermal stress

## 1. Introduction

### 1.1. Background

Low-temperature cracking of asphalt pavements can be a serious problem in regions where the temperature drops to extremely low values, such as −20 °C or lower. When the pavement is cooled to a temperature significantly lower than 0 °C, tensile stresses develop in the asphalt layer as a result of the pavement's tendency to contract. During winter, low-temperature cracks develop at the surface of the pavement when tensile thermal stress induced in the asphalt layer during cooling equals and exceeds the tensile strength of the material [1–6]. Under repeated temperature cycles, the crack can penetrate to the full depth of the asphalt layers. As a consequence, the existence of transverse cracks caused by extremely low temperature leads to degradation of the pavement structure by water entering through the cracks. According to the literature, the addition of additives such as sulfur [7], bio-agents [8], rubber-bitumen granulate [9] or composition of polymer-rubber modified bitumen [10] can improve the low-temperature performance of asphalt mixtures. Zhao et al. [11] assessed the effect of mineral fiber addition on the bending creep test results at low temperatures. It was concluded that

adding mineral fiber increased the creep rate of asphalt mixtures, and low-temperature properties were improved. The problem of thermal stresses is important not only in asphalt layers but also in concrete pavements subjected to seasonal and daily changes of temperature [12,13].

Tensile thermal stresses that occur in the asphalt layers during cooling are very difficult to measure directly in the pavement structure. Therefore, correct analytical estimation of thermal stresses is of crucial importance for evaluation of asphalt binders and design of asphalt mixtures. Calculation of tensile thermal stresses in asphalt pavements has been a subject of research for over 50 years. Viscoelastic solutions for thermal stresses were developed in the early 1960s by Muki and Sternberg [14], Lee and Rogers [15] and Humpreys and Martin [16]. This approach was dedicated to a specific class of polymers and used the temperature-dependent viscoelastic material parameters. The viscoelastic solution was also adopted in 1965 by Monismith et al. [17]. The basic viscoelastic equation for thermal stress calculation is as follows:

$$\sigma(t) = \int_0^t E(t - \xi) \frac{\delta\varepsilon(\xi)}{\delta\xi} d\xi \tag{1}$$

where: $\sigma(t)$–thermal stress in a viscoelastic slab at time t, $E(t)$–relaxation modulus of the viscoelastic slab as a function of time t, $\varepsilon$ – thermal strain induced in the viscoelastic slab by change of temperature and calculated as:

$$\varepsilon(\xi) = \alpha_T[T(\xi) - T_0] \tag{2}$$

where: $\alpha_T$–linear coefficient of thermal expansion, $\xi$–reduced time associated with time and temperature: $\xi = \frac{t}{a_T}$ and $a_T$–temperature shift factor, $T(\xi)$–pavement temperature at reduced time $\xi$, $T_0$–pavement temperature when $\sigma(t) = 0$.

The viscoelastic solution presented in Equations (1) and (2) has been used for calculation of thermal stresses in asphalt layers by several researchers [18–25]. This method has also been incorporated in the new AASHTO mechanistic-empirical method of pavement design [26] as well as the AASHTO PP 42-02 [27] and ASTM D6816-11 [28] standards.

In 1966, Hills and Brien introduced a simple quasi-elastic solution for calculation of thermal stresses [29]. Due to the fact that asphalt mixtures are viscoelastic materials with time-temperature dependent properties, the basic disadvantage of this method is that the viscoelastic nature of bituminous material and stress relaxation behavior is not fully considered. Nevertheless, mainly for its simplicity, the Hills and Brien method of thermal stress calculation has been used in several research papers [30–34]. In this method the thermal stresses are calculated from the following equation:

$$\sigma(T) = \frac{1}{1 - \nu} \sum_{i=1}^n S(t,\ T_i) \cdot \alpha_T \cdot \Delta T \tag{3}$$

where: $\sigma(T)$–thermal stress induced in an asphalt layer at temperature T, $\nu$–Poisson's ratio of the asphalt layer, $S(t,\ T)$–stiffness modulus of the asphalt layer as a function of time of loading $t$ and temperature $T_i$, $\alpha_T$–coefficient of thermal expansion of the asphalt layer, $\Delta T$–increment of temperature assumed in calculations, $i = 1, \ldots, n$–steps in calculations, $T_i$ temperature at step $i$ and $T_i = T_{i-1} + \Delta T$.

Thermal stresses can be also assessed in laboratory and the results of thermal stress calculations can be then compared to the laboratory test results. In the research of Qian et al. [35], thermal viscoelasticity theory was applied to simulate the Thermal Stress Restrained Specimen Test (TSRST). Gajewski and Langlois [36] modeled the TSRST results using the finite element method in a frame of thermo-mechanics with a "weak coupling" between thermal and mechanical effects. As an alternative to the TSRST method, the Asphalt Concrete Cracking Device (ACCD) was developed by Akentuna et al. to study thermal stress development in asphalt mixtures [37]. Yavuzturk and Ksaibati [38] developed a computer model using a transient, two-dimensional finite volume approach to mathematically describe the thermal response of asphalt pavements to thermal environmental conditions on an hourly basis. Creep tests are the most common tests used to assess low-temperature properties of asphalt mixtures in laboratory conditions [39–43]. Thermal stresses can be also calculated on the basis of test results of

asphalt mixture samples cored from existing pavement structures [44]. The innovative nature of this paper is the application of the tensile creep test procedure and its evaluation for the assessment of low temperature properties. The nature of thermal stresses that are built up in the asphalt pavement when the temperature decreases is tension rather than bending. Most of the research projects concerning thermal stresses calculation are based on bending tests. A more detailed study of tensile behavior and its comparison with previous test results based on three point bending test seems to be important.

*1.2. Objectives*

The main objective of the paper is to assess the influence of creep test methods on the obtained results of low-temperature properties of asphalt mixtures, especially the thermal stresses induced in asphalt pavement by a decrease in temperature during winter. For this purpose, three point bending and uniaxial tensile creep tests were applied and the master curves of stiffness modulus were analyzed. Rheological parameters describing elastic and viscous properties of the asphalt mixtures were determined on the basis of creep test results. Burgers' rheological model was used to calculate the rheological parameters. Thermal stresses were calculated and compared to tensile strength of the material to obtain the failure temperature of the analyzed asphalt mixtures.

## 2. Materials and Methods

*2.1. Materials*

### 2.1.1. Bitumen

Three types of neat road bitumen–35/50, 50/70 and 70/100, produced according to EN 12591 standard [45] and one polymer Styrene-Butadiene-Styrene modified bitumen 45/80-55, produced according to EN 14023 standard [46], were selected for the assessment of low-temperature creep properties of asphalt mixtures. Standard properties of the bitumen used in this research are shown in Table 1.

**Table 1.** Properties of bitumen.

| | | Type of Bitumen | | | |
|---|---|---|---|---|---|
| **Property** | | **35/50** | **50/70** | **70/100** | **45/80-55** |
| Penetration at 25 °C, 0.1 mm, | Original | 45 | 54 | 81 | 60 |
| acc. to PN-EN 1426 | RTFOT | 28 | 40 | 48 | 40 |
| R&B Temperature, °C, | Original | 53.0 | 50.8 | 47.8 | 68.6 |
| acc. to PN-EN 1427 | RTFOT | 57.8 | 57.8 | 53.4 | 67.4 |
| Performance Grade, acc. to AASHTO M 320 | | 70-16 | 64-22 | 58-22 | 70-22 |
| Fraass Breaking Point Temperature, °C, acc. to | Original | −6 | −14 | −16 | −16 |
| PN-EN 12593 | RTFOT | −3 | −12 | −10 | −15 |

### 2.1.2. Asphalt Mixtures

Laboratory tests were conducted on three types of asphalt mixtures: two wearing course asphalt concretes–AC 11 S for low traffic (LT) and AC 11 S for medium traffic (MT) – as well as one binder course asphalt concrete AC 11 W for medium traffic (MT). All mixes were designed in compliance with the EN 13108-1 standard [47] and were prepared in the laboratory [48]. The mixture type AC 11 S (MT) was designed in two variants (using the neat 50/70 bitumen and the modified 45/80-55 bitumen), therefore, a total of four different asphalt mixtures was used in the test. The compositions of mixtures and types of bitumen used are presented in Table 2.

**Table 2.** Properties of asphalt mixtures.

| Asphalt Mixture | Type of Mixtures | | |
| --- | --- | --- | --- |
| | **AC 11 S** | **AC 11 S** | **AC 11 W** |
| Type of layer | wearing course | wearing course | binder course |
| Type of traffic | low traffic (LT) | medium traffic (MT) | medium traffic (MT) |
| Bitumen type | 70/100 | 50/70 45/80-55 | 35/50 |
| Binder content (% by mass) | 5.8 | 5.6 | 5.6 |
| Aggregate type | crushed gravel | crushed gneiss | crushed gneiss |
| Filler type | limestone | limestone | limestone |
| Sieve size (mm) | % Passing (by mass) | | |
| 16 | 100 | 100 | 100 |
| 11.2 | 97 | 98 | 98 |
| 8 | 83 | 77 | 83 |
| 5.6 | 71 | 62 | 65 |
| 4 | 60 | 52 | 54 |
| 2 | 40 | 39 | 43 |
| 0.125 | 11 | 11 | 12 |
| 0.063 | 8.0 | 7.2 | 7.4 |

*2.2. Methods*

2.2.1. Tensile Creep Test (TCT)

Tensile creep properties of asphalt mixtures at low temperatures were assessed by means of the Tensile Creep Test (TCT) method according to EN 12697-46 standard [49]. In the TCT, the specimen is subjected to a constant tensile stress $\sigma$ at a constant temperature $T$. The progression of the strain $\varepsilon$ with time is recorded. According to the standard, it is recommended to maintain the constant load for 8 h and, after unloading, record the regression for additional 2 h. The principle of the TCT is shown in Figure 1.

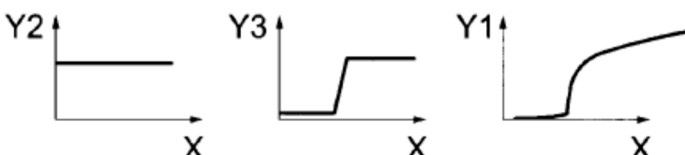

**Figure 1.** Principle of the Tensile Creep Test (TCT); where: **X**–time, **Y1**–strain, **Y2**–temperature, **Y3**–stress, [49].

In the TCT method, the specimens were loaded in order to achieve a constant tensile stress at a constant temperature. The stress levels were determined in relation to tensile strength results from the Uniaxial Tension Stress Test (UTST). In the UTST, a specimen is pulled with a constant strain rate at a constant temperature until failure. Results of the UTST are: the maximum stress (tensile strength) $\beta_t(T)$ and the corresponding tensile failure strain $\varepsilon_{failure}(T)$ at the test temperature $T$. In this research, the Tensile Stress Restrained Specimen Test (TSRST) method was used as well. The results from the TSRST were compared to thermal stresses calculated based on the results from the creep test methods. The results of the TSRST procedure are: the progression of the thermal stress over the temperature $\sigma_{cry}(T)$ and the failure stress $\sigma_{cry, failure}(T)$ at the failure temperature $T_{failure}$. The results from the UTST and TSRST have been published and discussed in a separate paper [6]. In the TCT method, the level of the stresses applied was determined as the percentage of the maximum stress $\beta_t(T)$ from the UTST at a given test temperature. Temperatures and stresses applied in the TCT are presented in Table 3 and in Figure 2.

**Table 3.** Test conditions for the Tensile Creep Test (TCT), [49].

| Test Temperature $T$, °C | Percentage of the Maximum Stress $\beta_t(T)$ Obtained from the UTST, % |
|---|---|
| +20 | 5 |
| +5 | 10 |
| −10 | 30 |
| −20 | 50 |
| −30 | 50 |

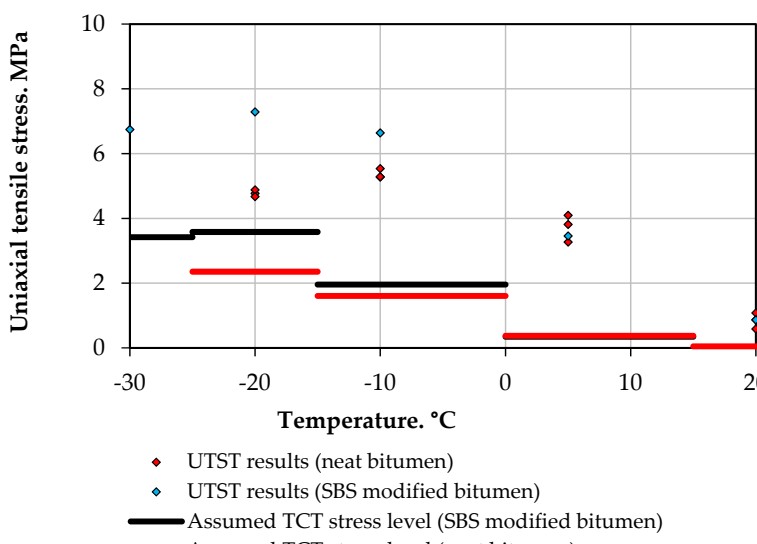

**Figure 2.** Stress levels at test temperatures applied in the Tensile Creep Test (TCT).

The specimens were tested using a "TSRST-MULTI" Multi Station Thermal Asphalt System (PAVETEST, Italy) device with servo hydraulic equipment. The equipment and test setup used are presented in Figure 3.

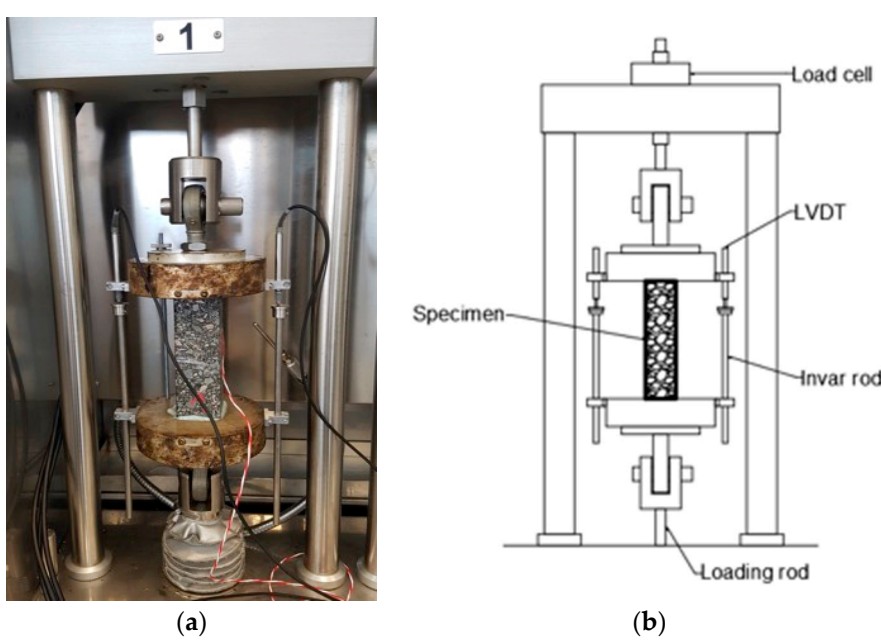

**Figure 3.** Tensile Creep Test (TCT) and Uniaxial Tension Stress Test (UTST) setup: (**a**) photograph of the specimens during the test, (**b**) schematic view.

Three specimens were tested for each asphalt mixture and test temperature (both in the TCT or the UTST). The specimens were sawn from slabs compacted in the laboratory according to EN 12697-33 [50], in order to obtain the specimen shape of prismatic beam with dimensions of 40 × 40 × 160 mm. The specimens were sawn from the middle of the slab with the distance of each specimen to the border being at least 20 mm. In the TCT and UTST procedures, the specimens were tested at constant test temperatures: −20 °C, −10 °C, +5 °C and +20 °C. In the case of asphalt mixture with SBS-polymer modified bitumen, the specimens were also tested at the temperature of −30 °C. The constant strain rate applied in the UTST was $\Delta\varepsilon$ = 0.625 ± 0.025 %/min, which corresponds to tension rate of 1 mm/min.

The TCT procedure comprises two main stages. In the first stage, the prismatic specimen is subjected to constant load for 28,800 s (8 h). In the second stage, after the specimen is unloaded, and its regression is recorded for 7200 s according to the EN 12697-46 standard [49].

### 2.2.2. Bending Beam Creep Test (BBCT)

The basic procedure of the Bending Beam Creep Test (BBCT) was developed by Judycki [51] and later improved and described by Pszczola et al. [39]. In the test at least 5 prismatic specimens (50 × 50 × 300 mm) are used for every test temperature. Specimens are sawn from plates (300 × 300 × 50 mm) of the asphalt mixture, compacted using a standard laboratory roller compactor. The degree of compaction is equal to 99% of Marshall specimen bulk density. The BBCT was conducted at four temperatures: −20 °C, −10 °C, 0 °C and +10 °C. Before the test, each specimen was subjected to the temperature of the test for 12 h. A specimen mounted in the test equipment and its schematic view are presented in Figure 4.

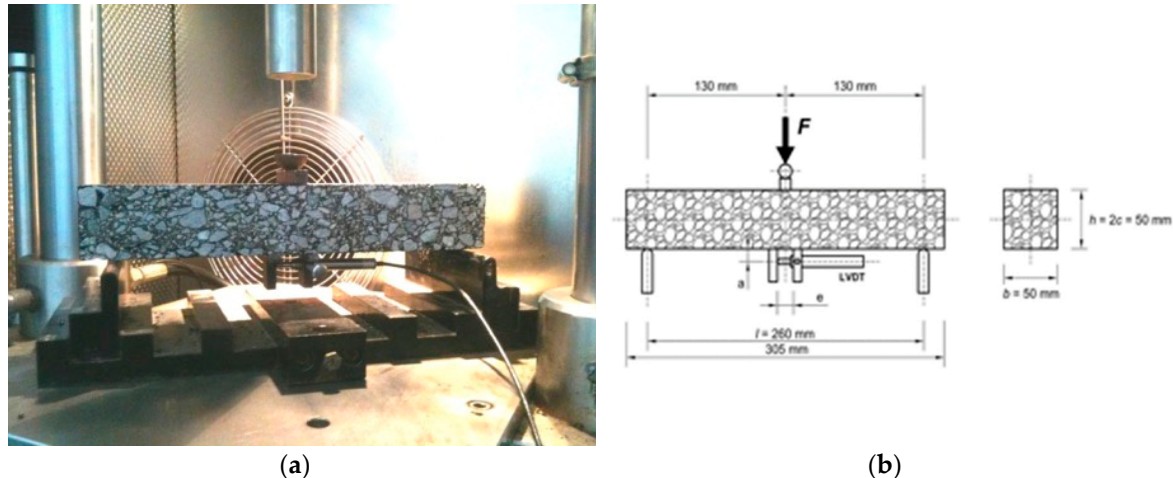

(**a**)　　　　　　　　　　　　　　　　　　　　　　　　　(**b**)

**Figure 4.** Bending Beam Creep Test: (**a**) specimen mounted in the Bending Beam Creep tester; (**b**) schematic view [39].

Stress levels and test temperatures in the Bending Beam Creep Test are presented in Figure 5.

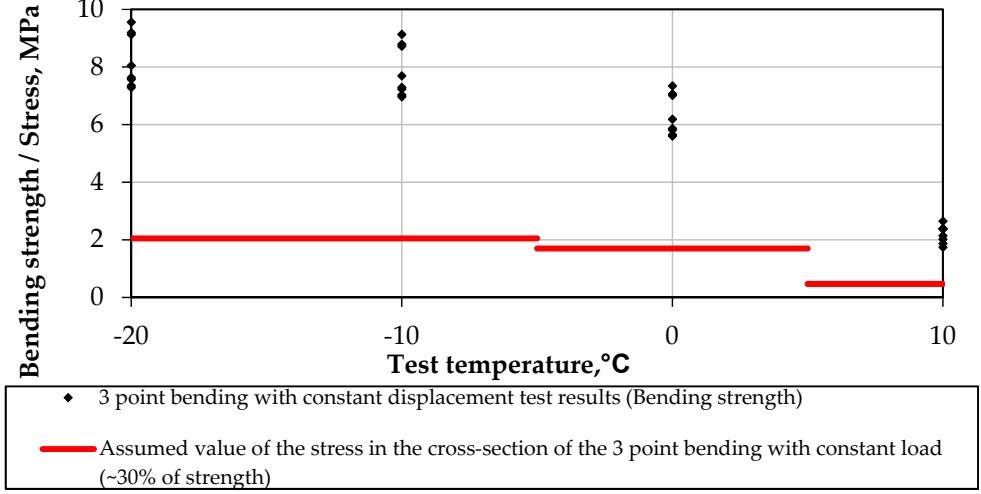

**Figure 5.** Stress levels at various test temperatures applied in the Bending Beam Creep Test (BBCT).

*2.3. Method of Calculation of Rheological Properties*

On the basis of TCT and BBCT results, rheological parameters describing elastic and viscous properties of the asphalt mixtures were determined. Burgers' rheological model was used to calculate the rheological parameters that were later utilized for calculation of thermal stresses. Burgers' model parameters are determined using the least square method. For this purpose, each of the creep curves is described using Equation (4), where Burgers' model parameters are treated as fitting parameters.

$$\varepsilon(T,t) = \sigma_0 \cdot \left\{ \frac{1}{E_1} + \frac{t}{\eta_1} + \frac{1}{E_2} \left[ 1 - e^{\left(\frac{-t}{\lambda}\right)} \right] \right\} \tag{4}$$

where: $\lambda = \eta_2/E_2$, $E_1$—instantaneous modulus of elasticity, MPa; $E_2$—modulus of retarded elasticity, MPa; $\eta_1$—coefficient of viscosity of steady flow, MPa·s; $\eta_2$—coefficient of viscosity of retarded flow, MPa·s; $t$—time of loading, s, $\sigma_0$—constant stress during load phase, specific for each temperature, MPa.

Under assumption of the time-temperature superposition principle [52], on the basis of TCT and BBCT results obtained for 4 different temperatures (either +20 °C, +5 °C, −10 °C, −20 °C or +10 °C, 0 °C, −10 °C, −20 °C), the master curves of stiffness modulus were developed using Richards model [53], which is given by Equation (5):

$$log|E^*| = \delta + \frac{\alpha - \delta}{\left[1 + \lambda e^{\beta + \gamma log f}\right]^{(1/\lambda)}} \tag{5}$$

: $log|E^*|$—stiffness modulus, psi; $f$—reduced frequency, Hz; $\alpha$, $\delta$, $\beta$, $\gamma$, $\lambda$—master curve fitting parameters. Reference temperature was selected as −10 °C.

## 3. Results and Discussion

Bending and tensile creep tests conducted in this study posed a base for determination of various rheological properties of the tested asphalt mixtures. Afterward, selected properties were used in calculation of thermal stresses developed in the pavement due to a decrease in temperature. The following characteristics were determined from the conducted laboratory tests: stiffness moduli, Burgers' model parameters and master curve equations with appropriate shift factors. All the derived basic properties were assessed taking into assumption linear viscoelasticity and thermo-rheological simplicity. In the case of master curves, the Richards "branching" model modification [54] properties were determined as well.

### 3.1. Stiffness Modulus

Stiffness curves derived from the results of the two creep tests—the Bending Beam Creep Test (BBCT) and the Tensile Creep Test (TCT)—for all the tested materials are presented in Figures 6–9. In the case of the Bending Beam Creep Test, the presented results are the mean values from 5 different specimens (coefficient of variation for all test results is in the range from 5 to 25%; mean 10%). In the case of the Tensile Creep Test, the presented results are the mean values from 2 to 4 different specimens (coefficient of variation for all test results is in the range from 1 to 40%; mean 15%). For the asphalt mixture AC 11W with neat bitumen 35/50, only the Tensile Creep Test was conducted.

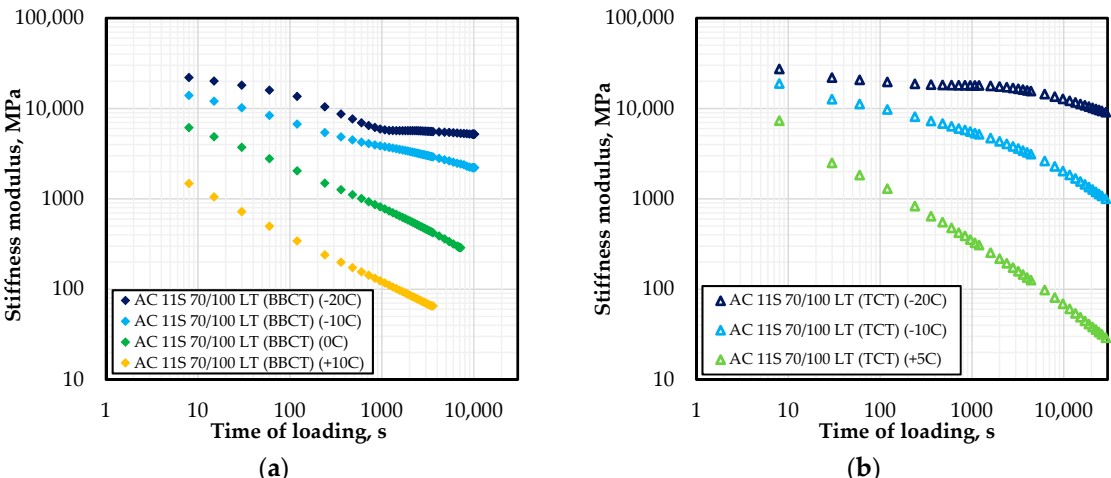

**Figure 6.** Creep test results—stiffness curves for AC 11S 70/100 LT (**a**) BBCT; (**b**) TCT.

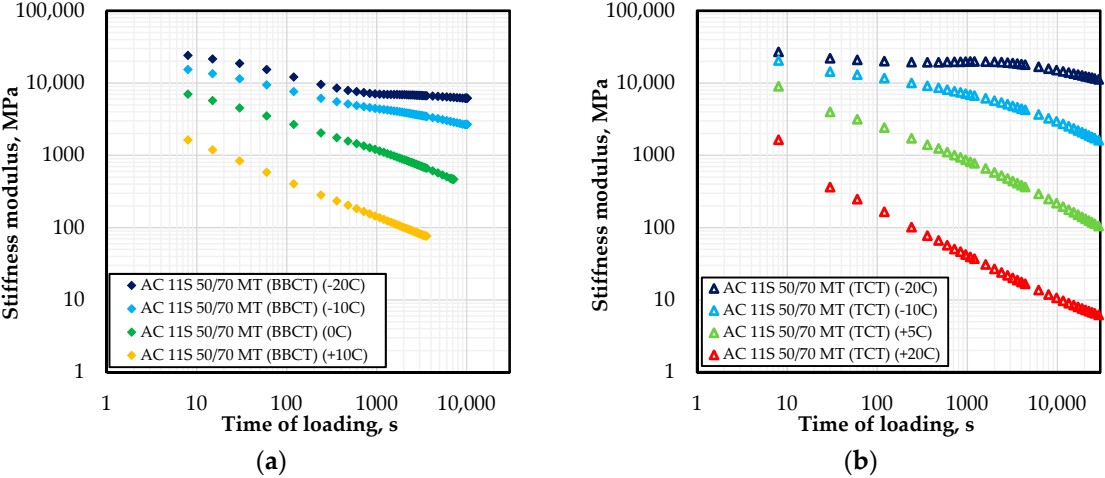

**Figure 7.** Creep test results—stiffness curves for AC 11S 50/70 MT (**a**) BBCT; (**b**) TCT.

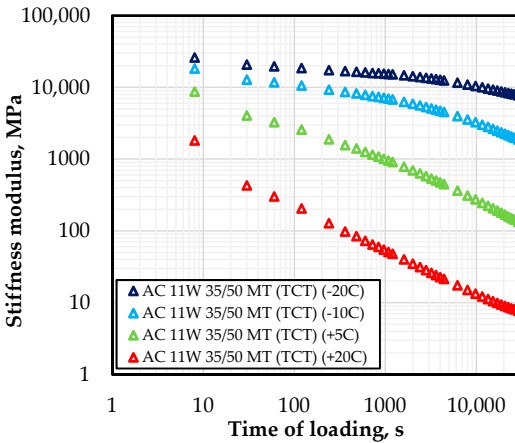

**Figure 8.** Creep test results—stiffness curves for AC 11W 35/50 MT, only for TCT.

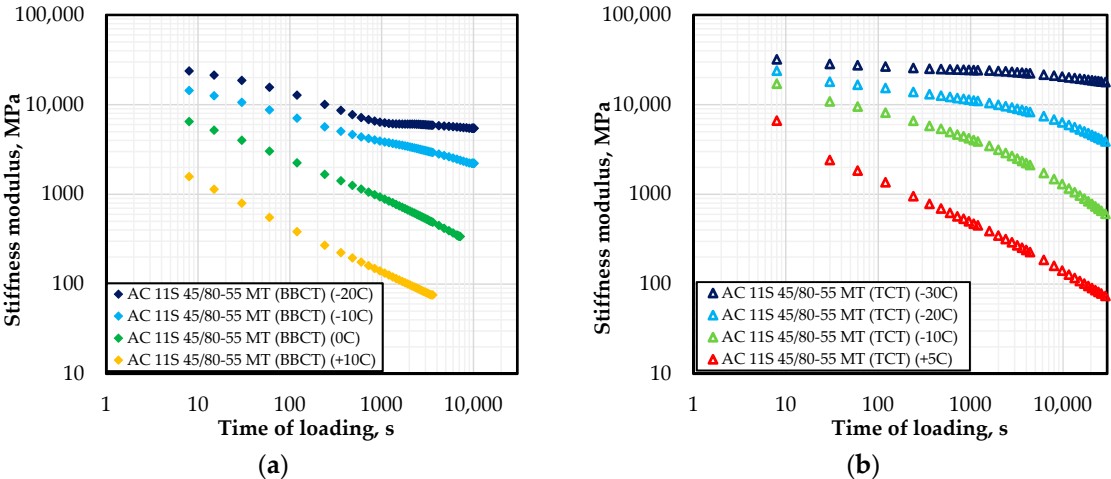

**Figure 9.** Creep test results—stiffness curves for AC 11S 45/80-55 MT (**a**) BBCT; (**b**) TCT.

Stiffness curves derived from both creep tests differ at all the tested temperatures—both by shape and by values. For low temperatures (−10 °C and lower) results obtained from the TCT present higher values of stiffness modulus. The situation is opposite at temperatures higher than 0 °C, where results obtained from the BBCT present higher values. The difference in both cases is around 50% and will be discussed in detail further in the paper.

In terms of shape of the stiffness curves, two main differences are visible between the two tests: the shape of the stiffness curve at the highest temperature as well as the occurrence and shape of deviations from the time-temperature superposition principle. As shown in Figures 6b and 7b, for the temperature of +20 °C the stiffness curve derived from the TCT bends from a straight line for long times of loading. Such a shape is beneficial for construction of the master curve—it is easier to develop the equation for a typical sigmoidal shape, where "Max" and "$\delta$" master curve parameters could have physical meaning of maximum and minimum stiffness modulus. In the case of BBCT test trials at +20 °C for the lowest possible stress, the results suggested destruction of the specimen for times of loading longer than 200 s (stiffness moduli reached value of ~1 MPa). Therefore, in the case of BBCT, it is much harder to derive the "$\delta$" master curve parameter, which corresponds to the lowest value. As for the appearance of deviations—previous studies [40,54] showed that in the case of BBCT test, deviations appeared after around 500 s of loading. Their shape suggested that the curve was approaching a horizontal asymptote. Such results were confirmed in other studies [10,39]. In the case of the TCT test, deviations from the time-temperature superposition principle were also observed for most cases (except for the AC 11S 45/80-55 MT mixture) at temperatures of −10 °C and lower. However, first analyses showed that they occur earlier (around 200-300 s) and the shape

is different—after the "asymptotical" phase, which lasts around 1000-3000 s, the modulus starts to decrease again. As further tests will be conducted on other mixtures, deviations will be analyzed in detail in future studies.

### 3.2. Burgers' Model Parameters

Burgers' model parameters were derived using the procedure presented in Section 2.3. The procedure was the same for creep curves determined in both tests. Results for the TCT are presented in Table 4 and results for the BBCT are presented in Table 5.

**Table 4.** Burgers' model parameters for the Tensile Creep Test (TCT)

| Mixture Designation | Test Temperature T, °C | Burgers Model Parameters | | | |
|---|---|---|---|---|---|
| | | $E_1$, MPa | $E_2$, MPa | $\eta_1$, MPa·s | $\eta_2$, MPa·s |
| AC 11S 70/100 LT | −20 | 24,601 | 60,649 | 620,381,505 | 83,584,308 |
| | −10 | 12,516 | 6790 | 38,457,749 | 14,471,875 |
| | 5 | 2791 | 254 | 991,934 | 767,794 |
| | 20 | * | * | * | * |
| AC 11S 50/70 MT | −20 | 24,656 | 115,183 | 827,458,047 | 90,290,579 |
| | −10 | 13,737 | 7758 | 69,265,928 | 21,096,067 |
| | 5 | 4070 | 459 | 4,086,935 | 1,888,343 |
| | 20 | 629 | 14 | 353,051 | 62,056 |
| AC 11S 45/80-55 MT | −30 | 29,941 | 98,166 | 2,381,475,542 | 91,889,789 |
| | −20 | 19,307 | 19,490 | 195,406,816 | 28,798,508 |
| | −10 | 12,257 | 4123 | 21,890,814 | 10,897,801 |
| | 5 | 2762 | 239 | 3,205,784 | 1,009,562 |
| AC 11W 35/50 MT | −20 | 21,772 | 28,228 | 600,059,352 | 75,959,661 |
| | −10 | 13,004 | 8210 | 87,034,368 | 23,417,485 |
| | 5 | 4031 | 601 | 5,187,255 | 2,240,592 |
| | 20 | 749 | 16 | 453,536 | 85,500 |

*–specimen failure during the test.

**Table 5.** Burgers' model parameters for the Bending Beam Creep Test (BBCT).

| Mixture Designation | Test Temperature T, °C | Burgers' Model Parameters | | | |
|---|---|---|---|---|---|
| | | $E_1$, MPa | $E_2$, MPa | $\eta_1$, MPa·s | $\eta_2$, MPa·s |
| AC 11S 70/100 LT | −20 | 38,503 | 4,517 | 588,962,264 | 2,007,996 |
| | −10 | 34,090 | 2,948 | 60,779,274 | 624,963 |
| | 0 | 18,572 | 801 | 3,152,349 | 271,364 |
| | 10 | 6,168 | 151 | 397,640 | 41,598 |
| AC 11S 50/70 MT | −20 | 46,106 | 5,024 | 595,720,702 | 470,341 |
| | −10 | 34,208 | 3,350 | 79,945,328 | 490,828 |
| | 0 | 18,582 | 1,153 | 5,340,236 | 303,322 |
| | 10 | 5,129 | 175 | 467,638 | 49,697 |
| AC 11S 45/80-55 MT | −20 | 45,953 | 4,676 | 487,685,841 | 588,195 |
| | −10 | 30,444 | 3,149 | 53,177,748 | 437,444 |
| | 0 | 17,695 | 876 | 3,849,752 | 319,700 |
| | 10 | 5,134 | 167 | 478,026 | 47,128 |

Burgers' model parameters differ in all cases, apart from the $\eta_1$ coefficient of viscosity steady flow, in whose case values are comparable. Secondary parameters $E_2$ and $\eta_2$ present higher values for the TCT and show better consistency with the results determined from dynamic tests [55]. In the case of the $E_1$ modulus the situation is more complicated. While the values obtained from the TCT are almost two times lower than those derived from the BBCT and seem more realistic, a reduction in values obtained from the TCT could have been caused to some extent by the less accurate record of the

creep curve in the unloading part. The first record is made after 25 s. Nevertheless, the results of the $E_1$ Burgers' model parameter derived from the TCT show more consistency with the stiffness curves presented in Section 3.1. The highest values of stiffness modulus for the temperatures of $-10\ °C$ and $-20\ °C$ are similar for both tests.

### 3.3. Master Curve Parameters

On the basis of previous studies and literature review [10,39,40,52,53,56], Richards model was selected as the basic equation of master curves. "Branching" model modification [54] was also applied for all the tested mixtures, to take into account the deviations from the time-temperature superposition principle. Parameters for asphalt mixtures derived from the TCT are presented in Table 6. Parameters for asphalt mixtures derived from the BBCT are presented in Table 7. Shift factors used in the study are presented in Figure 10.

Basic parameters of the master curve described by the Richards model in which deviations are completely omitted are presented in the first row of results for each mixture (for $T > -20\ °C$ or $T > -10\ °C$, depending on the mixture and test procedure). It is visible that in the case of higher temperatures for the TCT, the "$\delta$" parameter shows consistency between all the tested mixtures, and it is always higher than 0. Determination of the same parameter for the BBCT sometimes requires setting of the minimum value of "$\delta$" as 0, as the SOLVER unit of EXCEL used in the analysis sometimes suggests values lower than 0 [40], which is in contradiction to the physical meaning of the parameter. Additional "branches", deviating from the master curves due to shifting of stiffness curves obtained at lower test temperatures, are presented in further rows of the tables, according to their corresponding temperature ranges. Since the shape of the deviations in the TCT is different than in the BBCT, the values of the "$\delta$" parameter for the TCT are equal across all temperatures. The "asymptotical" phase is not correctly described, which suggests that for the TCT a new model needs to be developed. Results presented in this study for the BBCT show full consistency with the "branching" modification of Richards model [54], and deviations are correctly described.

**Table 6.** Master curve parameters for the Tensile Creep Test (TCT).

| Mixture Designation | Test Temperature T, °C | Richards Model Parameters ("Branching" Modification) | | | | |
| --- | --- | --- | --- | --- | --- | --- |
| | | Max | β | γ | δ | λ |
| AC 11S 70/100 LT | $> -20$ | 4.712 | $-10.075$ | $-0.462$ | 0.367 | 0.000615 |
| | $< -20$ | 4.712 | $-17.217$ | $-0.279$ | 0.367 | 0.0000005 |
| AC 11S 50/70 MT | $> -20$ | 4.789 | $-9.735$ | $-0.451$ | 0.441 | 0.000898 |
| | $< -20$ | 4.789 | $-16.478$ | $-0.196$ | 0.441 | 0.000001 |
| AC 11S 45/80-55 MT | $> -20$ | 4.823 | $-9.584$ | $-0.421$ | 1.175 | 0.000566 |
| | $-20 \div -30$ | 4.823 | $-10.698$ | $-0.366$ | 0.823 | 0.000226 |
| | $< -30$ | 4.823 | $-10.424$ | $-0.212$ | 3.786 | 0.000066 |
| AC 11W 35/50 MT | $> -10$ | 4.935 | $-9.703$ | $-0.369$ | 0.284 | 0.00055 |
| | $-10 \div -20$ | 4.935 | $-6.737$ | $-0.302$ | 0.284 | 0.01074 |
| | $< -20$ | 4.935 | $-11.818$ | $-0.213$ | 0.284 | 0.00006 |

Two models of shift factors commonly used for binders and asphalt mixtures were used in the study—WLF and Arrhenius [52]. Arrhenius model parameters are similar for both tests, but results obtained from the BBCT presented higher homogeneity. In both cases the results were similar, both in shape and in values, to the WLF function determined for the TCT (Figure 10b). The WLF function proved more reliable when applied to the TCT results than to the BBCT results. For the BBCT test, in the case of higher temperatures, the WLF function (Figure 10a) presents a significantly different shape. While in the case of temperatures of up to $0\ °C$, the course of the line is similar to the TCT test, in the case of $+10\ °C$ the values decrease strongly, probably due to limitation of the BBCT test. It was impossible to obtain reliable results at temperatures higher than $+15\ °C$.

**Table 7.** Master curve parameters for the Bending Beam Creep Test (BBCT).

| Mixture Designation | Test Temperature T, °C | Richards Model Parameters ("Branching" Modification) | | | | |
|---|---|---|---|---|---|---|
| | | Max | β | γ | δ | λ |
| AC 11S 70/100 LT | >−10 | 4.766 | −9.066 | −0.428 | 1.327 | 0.000864 |
| | −10 ÷ −20 | 4.766 | −9.560 | −0.498 | 3.246 | 0.000206 |
| | <−20 | 4.766 | −10.471 | −0.848 | 3.713 | 0.000066 |
| AC 11S 50/70 MT | >−10 | 5.118 | −11.162 | −0.295 | 0.146 | 0.000090 |
| | −10 ÷ −20 | 5.118 | −11.835 | −0.368 | 3.234 | 0.000014 |
| | <−20 | 5.118 | −12.737 | −0.703 | 3.815 | 0.000004 |
| AC 11S 45/80-55 MT | >−10 | 4.764 | −10.941 | −0.429 | 1.358 | 0.000132 |
| | −10 ÷ −20 | 4.764 | −11.643 | −0.482 | 3.209 | 0.000026 |
| | <−20 | 4.764 | −11.381 | −0.852 | 3.741 | 0.000025 |

(**a**)

(**b**)

**Figure 10.** Shift factor—WLF function: (**a**) BBCT; (**b**) TCT.

Analysis of the master curve and shift factor models suggests that more reliable results should be obtained for the TCT, regardless of the model used. Some difficulties occurred for the "branching" modification of the Richards model, but further work should eliminate this problem. In the case of the BBCT, reliable results should be obtained using only Arrhenius shift factor model. The WLF model can influence analyses conducted at higher temperatures.

*3.4. Comparison of the TCT and BBCT Results*

Relationships between different test modes are commonly used in analyses of solid materials such as cement concrete. While Young's modulus should have the same value regardless of the test mode, the relationship between strength determined in simple tension and flexural test is approximately 1:2. While asphalt mixture strength relationships between different test modes were the subject of other studies [6,57], in this study, the authors focused on relationships between stiffness moduli. In Figures 11–14, the results obtained from both tests are compared in various forms: stiffness curves for chosen temperatures, shifted stiffness curves as well as master curves.

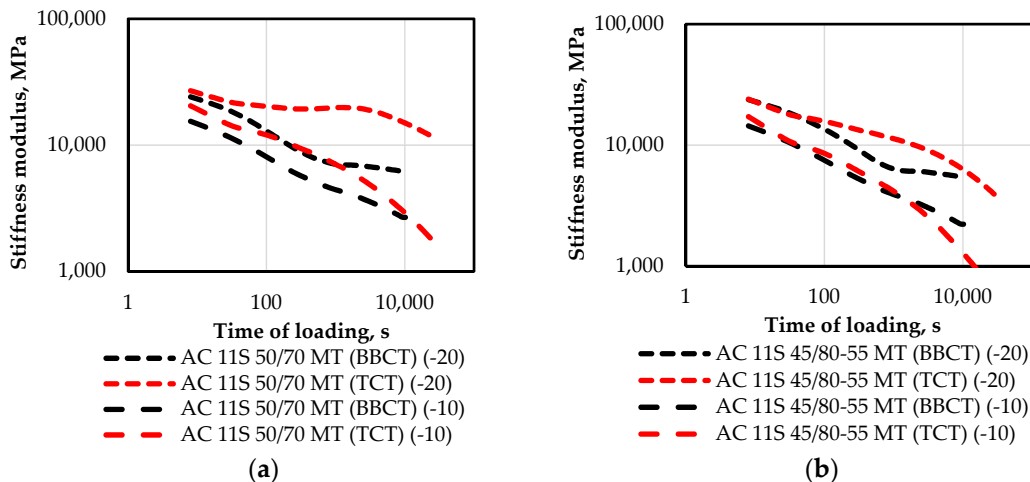

**Figure 11.** Comparison of stiffness curves at temperatures of −10 °C and −20 °C.

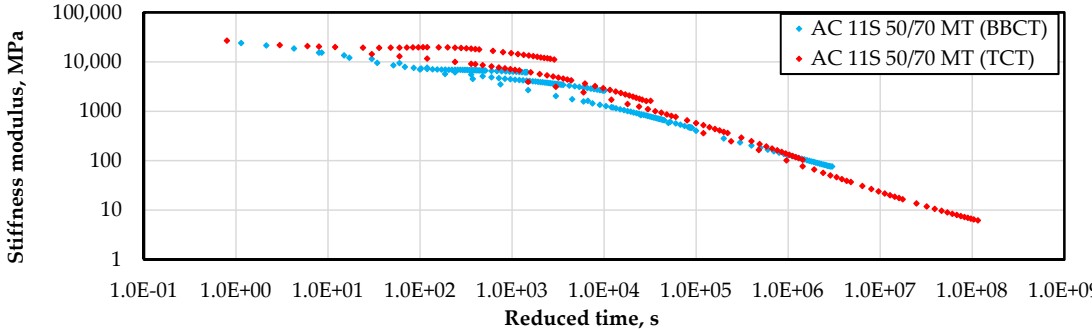

**Figure 12.** Comparison of shifted stiffness curves for AC 11S 50/70 MT, reference temperature: −10 °C.

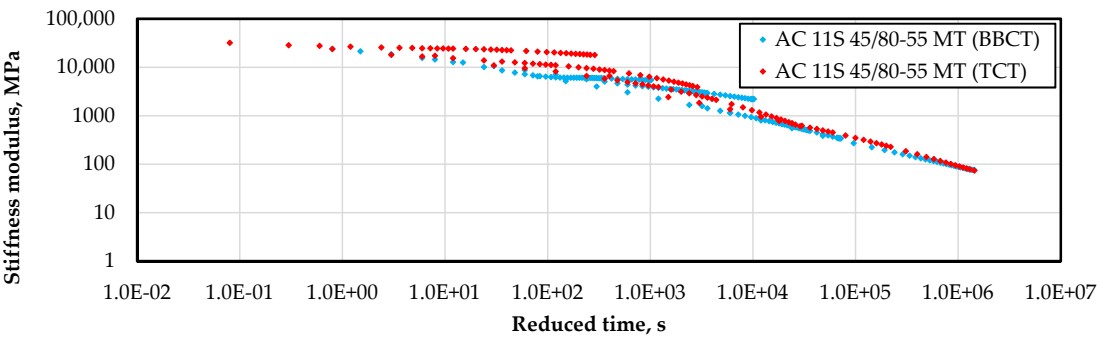

**Figure 13.** Comparison of shifted stiffness curves for AC 11S 45/80-55 MT, reference temperature: −10 °C.

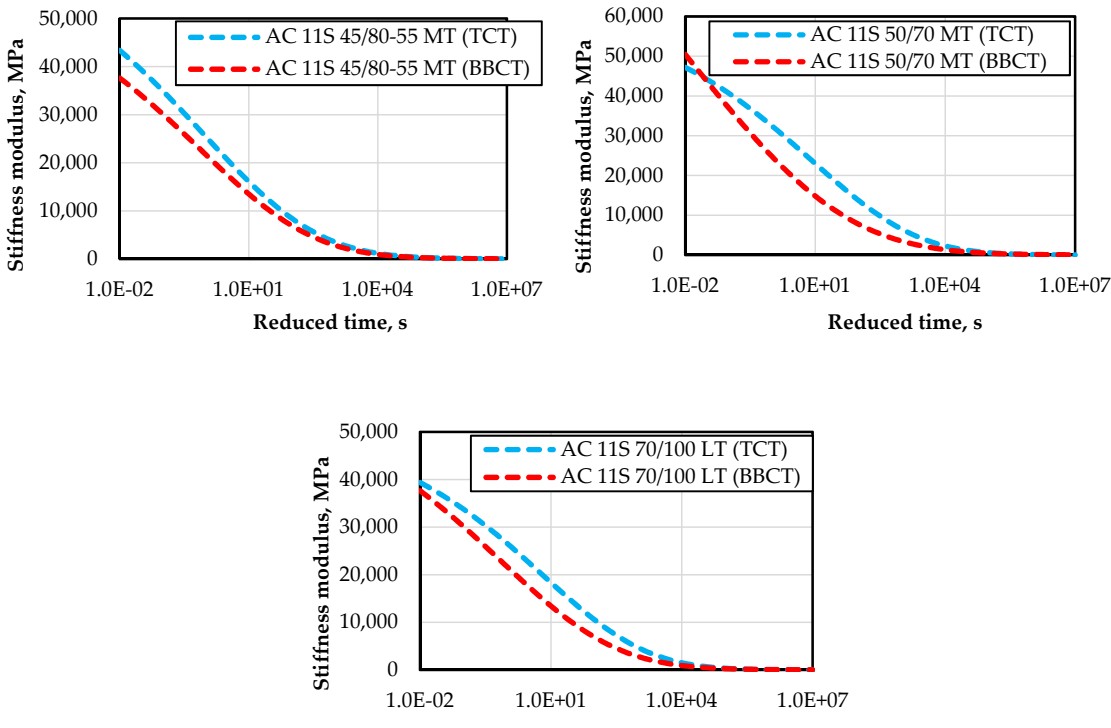

**Figure 14.** Comparison of master curves for all tested mixtures, reference temperature −10 °C.

As shown in Figure 11, for the first 100 s the results obtained from both tests are similar, with higher values of stiffness modulus obtained from the TCT. For longer times of loading the results differ strongly, both in values and the shape of the curve. The stiffness ratio between the TCT and the BBCT for the first 100 s is around 1.5 (Figure 15b). In the case of longer times of loading, the relationship is in the range from 1.1 to 4.0, with no visible tendencies for specific types of bitumen. Comparison of the shifted stiffness curves (Figures 12 and 13) shows that for lower temperatures (<0 °C) higher stiffness values are obtained for the TCT. The relationship changes at temperatures higher than 0 °C, for which the values of stiffness modulus obtained in the BBCT are higher. The same is visible in the developed master curves (Figure 14), where the master curve derived from the BBCT has a "flatter" shape than the one from the TCT. The stiffness ratio between master curves is presented in Figure 15a. While the range of variability is lower (between 0.5 and 2.0), no direct relationships are visible in this case, as opposed to the case of stiffness curves for shorter times of loading. For the majority of the reduced time, higher values are obtained for the TCT.

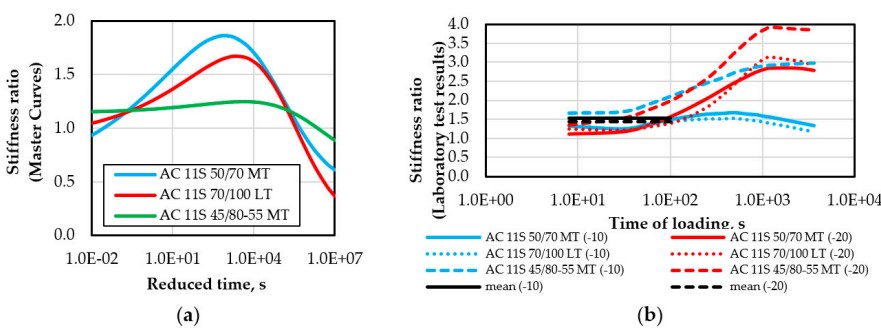

**Figure 15.** Stiffness ratio (TCT versus BBCT): (**a**) master curves (**b**) stiffness curves at temperatures of −10 °C and −20 °C.

## 4. Thermal Stress Analysis

Thermal stresses that developed due to a decrease in temperature were calculated using procedures presented in Section 1.1. For comparison with the TSRST [49], the gradient of temperature was assumed as 10 °C/h. For the Hills and Brien method, values of stiffness moduli were derived from stiffness curves as described in [5]. For the Monismith method, "branching" modification of the Richards model was used (shift factor – according to the Arrhenius model). In the case of AC 11W 35/50 MT, only results from the TCT were used. Calculated values of thermal stresses are presented in Figure 16. The predicted failure temperatures determined from Figure 16 are presented in Table 8.

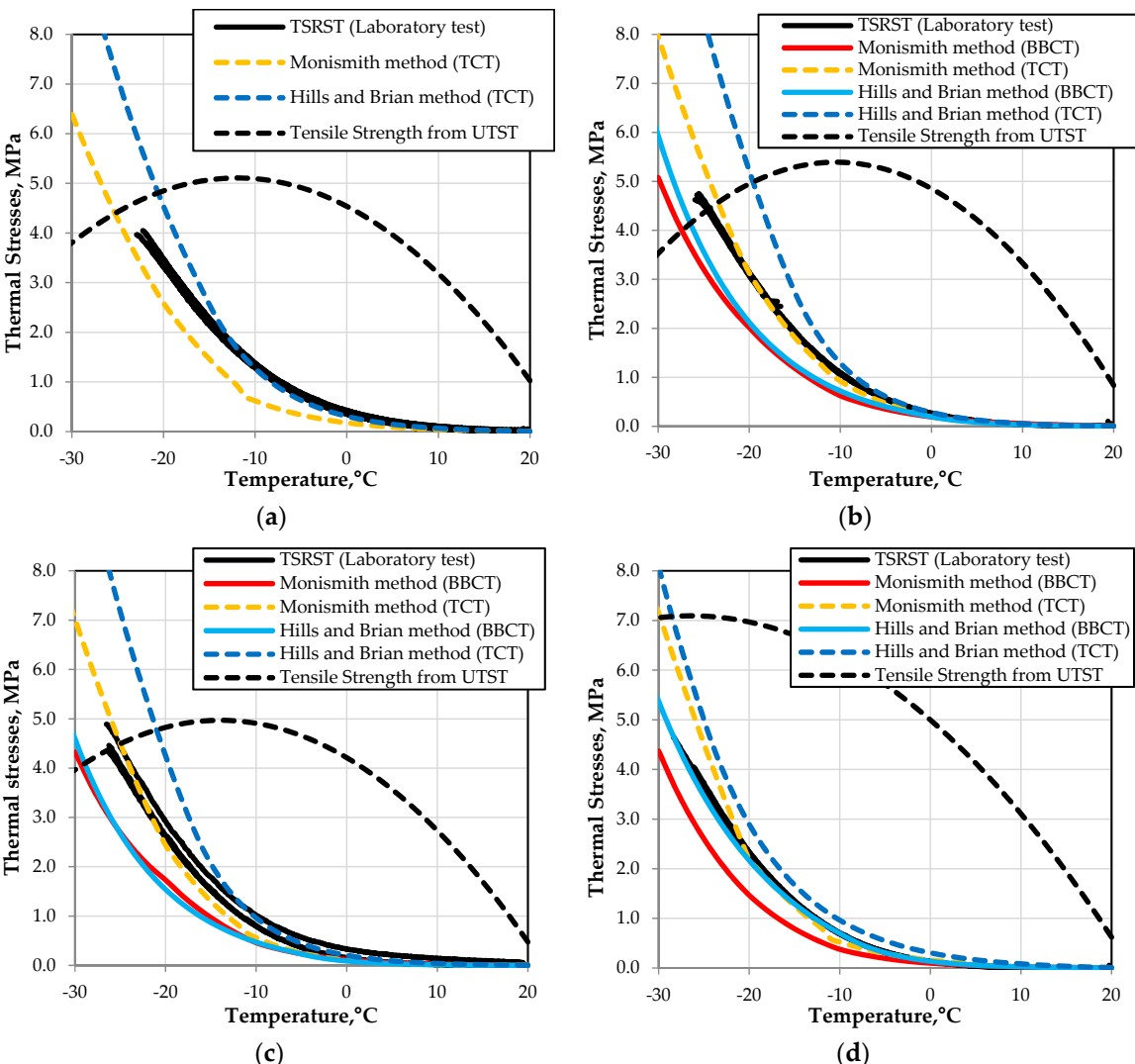

**Figure 16.** Results of calculation of thermal stresses for all the tested mixtures in comparison to the TSRST and UTST results, temperature gradient of 10 °C/h: (**a**) AC 11W 35/50 MT (**b**) AC 11S 50/70 MT (**c**) AC 11S 70/100 LT (**d**) AC 11S 45/80-55 MT.

As shown in Figure 16, the highest agreement with the TSRST results was obtained for the Monismith method based on the TCT results. In most cases, the TSRST results (black line) are almost the same. Lower values were often obtained for both thermal stress calculation methods (Hills & Brien as well as Monismith) when the BBCT results were used. In the case of the Hills & Brien method based on the TCT results, almost in all cases, the values of stress were higher than those obtained from the TSRST, which could result from the fact that stress relaxation is not fully considered in this method.

**Table 8.** Comparison of predicted failure temperatures (mean values).

| Mixture Designation | TSRST | Thermal Stress Calculation Method (Data Source) | | | |
|---|---|---|---|---|---|
| | | Hills & Brien (BBCT) | Hills & Brien (TCT) | Monismith (BBCT) | Monismith (TCT) |
| AC 11S 70/100 LT | −26.4 | −29.0 | −21.0 | −29.5 | −25.0 |
| AC 11S 50/70 MT | −25.7 | −26.5 | −19.5 | −27.5 | −23.5 |
| AC 11S 45/80-55 MT | −29.5 | <−30.0 | −28.5 | <−30.0 | −29.5 |
| AC 11W 35/50 MT | −22.3 | - | −20.5 | - | −25.5 |

In the course of assessment of failure temperatures presented in Table 8, no clear relationships were noted. The highest agreement was obtained for the Monismith method calculated based on the TCT results, for which 2 of 4 cases were the determined values of failure temperature in good agreement with the TSRST results.

The conducted analysis suggests that the highest reliability in calculations of thermal stresses was obtained for the Monismith method based on the TCT results. For the results of calculations based on the BBCT, a "method constant" should be determined in order to shift the results calculated from the BBCT into those calculated from the TCT, similar to the BBR methodology [27]. In this standard, to obtain values of thermal stresses for asphalt mixtures, one should multiply the thermal stresses calculated according to the AASHTO PP42-02 standard by a factor of 18. The presented study showed that such a factor for the BBCT results is in the range from 1.5 to 1.7, but it should be verified in more detailed studies encompassing other types of mixtures and bitumen.

## 5. Summary and Conclusions

This paper presents the study of low-temperature creep properties of asphalt mixtures and the methods of thermal stress assessment. Three point bending and uniaxial tensile creep tests were applied to calculate thermal stresses at low temperatures. Based on the test results and analysis, the following conclusions can be drawn:

1.  According to different creep test methods applied in the study, lower strain values of creep curves were obtained for the Tensile Creep Test (TCT) than for the Bending Beam Creep Test (BBCT), especially at lower temperatures.
2.  Stiffness curves derived from both creep tests (BBCT and TCT) differ at all the tested temperatures – both in terms of shape and values. For low temperatures (−10 °C and lower) results obtained from the TCT presented higher values of stiffness modulus. The situation is opposite at temperatures higher than 0 °C, where the results obtained from the BBCT present higher values.
3.  Master curves determined on the basis of the TCT results showed higher values of stiffness modulus for temperatures <0 °C and lower for temperatures >0°C in comparison to those determined from the BBCT.
4.  "Branching" modification of the Richards model correctly described the master curves determined from the BBCT results. In the case of the TCT results, master curves presented some discrepancies and a new model should be determined.
5.  The Arrhenius shift factor function presented reliable results for both tests (BBCT and TCT). In the case of the WLF shift factor function, the BBCT results showed significantly decreased values at temperatures higher than 0 °C.

6.   Results of thermal stress calculations indicated that higher reliability was obtained for the viscoelastic Monismith method based on the TCT results. The highest agreement with the TSRST results was also obtained for the Monismith method based on the TCT results.

7.   No clear relationships were obtained between the failure temperatures predicted from different methods of thermal stress calculation. The highest agreement was obtained for the viscoelastic Monismith method calculated based on the TCT results, for which 2 of 4 cases were the determined values of failure temperature in good agreement with the TSRST results.

8.   The main limitation of the study was related to the methodology of the TCT method. The test method was applied and conducted according to the European standard EN 12697-46. In the TCT, the time of loading is long (8 h) and an additional measurement period of 2 h is also recommended after unloading. The authors have come to the opinion that the time of loading can be shortened, but this issue requires further research.

**Author Contributions:** M.P. conceived and designed the experiments; C.S. and M.P. coordinated the experiments; M.P. and M.J. analyzed the TCT and BBCT data; M.P., M.J. and C.S. wrote the paper; M.P. coordinated and edited the paper.

**Acknowledgments:** The research was partly supported by the project RID-1B, financed by the National Center for Research and Development and the General Directorate for National Roads and Motorways under the program "Development of Road Innovations".

**Conflicts of Interest:** The authors declare no conflict of interest. The founding sponsors had no role in the design of the study; in the collection, analyses, or interpretation of data; in the writing of the manuscript, and in the decision to publish the results.

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
