# Peer review of "Assessment of Thermal Stresses in Asphalt Mixtures at Low Temperatures Using the Tensile Creep Test and the Bending Beam Creep Test"

_applsci, doi:10.3390/app9050846_

Round 1

Reviewer 1 Report

Thank you very much for this interesting paper on the subject of thermal stresses in asphalt concrete, which seems to be recently a subject of lower interest, and yet of such a high importance. I understand that this paper is presenting the experimental data supporting previously published paper by prof. Judycki (IJPE, 2018) in which a solution for the Burgers model was proposed as the most appropriate to describe rheological parameters in the material. Therefore the significance of this data and the article is of high improtance.

I appreciate the ease with which the authors are typically explaining complex problems, thus my request in respect of the Burgers model. Because I am a bit confused, sitting in between books during this review.

I have learnt about the model itself in the past being build on lambda as a parameter of the model. Lambda is often refered to as viscosity, but in fact it is viscous traction (as far as I know). In a way it is a viscosity but measured in one specific way during tensile loading. There is a relationship between viscosity (eta) and viscous traction (lambda), first studied by Couette.

I am at the moment sitting in front of 3 books dealing with rheology and few articles that present solutions, and unfortunately it seems that every author has their own take on the subject of eta and lambda for Burgers, and sometimes they are used as equal. Thus this creates a need for better referencing on hereby authors side to explain their chosen path or refer to the solution elsewhere.

In my opinion, if your experiment is measuring viscous traction then the viscosity value will be slightly different when equation is solved. As far as I can see TCT is measuring the viscous traction, but is the bending beam measuring the same? I think this is more of a shear situation for BBC. I think Burgers solution for both of them could be different, and this could be the reason why TCT is on some level making sense, but BBC seems to be giving odd results from the calculations for the retarded element? Perhaps the ratio of measured modulus has to do with that as well, the different viscosity measurement?

Otherwise, very interesting results and paper.

Language wise, the text between lines 230-253 has a surprising number of typos in comparison to the rest of the paper that seemed to be almost perfectly edited. I suggest revision of that chapter. And the correct word in English is "asymptote".

Author Response

We would like to express our thanks to the reviewer for the effort put into evaluation of our paper. The remarks have been taken into consideration and they will certainly improve the paper.

Comments and Suggestions of Reviewer 1:

“Thank you very much for this interesting paper on the subject of thermal stresses in asphalt concrete, which seems to be recently a subject of lower interest, and yet of such a high importance. I understand that this paper is presenting the experimental data supporting previously published paper by prof. Judycki (IJPE, 2018) in which a solution for the Burgers model was proposed as the most appropriate to describe rheological parameters in the material. Therefore the significance of this data and the article is of high importance.

I appreciate the ease with which the authors are typically explaining complex problems, thus my request in respect of the Burgers model. Because I am a bit confused, sitting in between books during this review.

I have learnt about the model itself in the past being built on lambda as a parameter of the model. Lambda is often referred to as viscosity, but in fact it is viscous traction (as far as I know). In a way it is a viscosity but measured in one specific way during tensile loading. There is a relationship between viscosity (eta) and viscous traction (lambda), first studied by Couette.

I am at the moment sitting in front of 3 books dealing with rheology and few articles that present solutions, and unfortunately it seems that every author has their own take on the subject of eta and lambda for Burgers, and sometimes they are used as equal. Thus this creates a need for better referencing on hereby authors side to explain their chosen path or refer to the solution elsewhere.

In my opinion, if your experiment is measuring viscous traction then the viscosity value will be slightly different when equation is solved. As far as I can see TCT is measuring the viscous traction, but is the bending beam measuring the same? I think this is more of a shear situation for BBC. I think Burgers solution for both of them could be different, and this could be the reason why TCT is on some level making sense, but BBC seems to be giving odd results from the calculations for the retarded element? Perhaps the ratio of measured modulus has to do with that as well, the different viscosity measurement?

Otherwise, very interesting results and paper.

Language wise, the text between lines 230-253 has a surprising number of typos in comparison to the rest of the paper that seemed to be almost perfectly edited. I suggest revision of that chapter. And the correct word in English is "asymptote".

Answers to the reviewer’s comments and suggestions. The authors would like to thank the reviewer for the general comments. Regarding the detailed suggestions, our answers are as follows:

1.     Thank you very much for this interesting paper on the subject of thermal stresses in asphalt concrete, which seems to be recently a subject of lower interest, and yet of such a high importance. I understand that this paper is presenting the experimental data supporting previously published paper by prof. Judycki (IJPE, 2018) in which a solution for the Burgers model was proposed as the most appropriate to describe rheological parameters in the material. Therefore the significance of this data and the article is of high importance.

Answer to comment 1: The authors would like to thank the reviewer for these comments.

2.     I appreciate the ease with which the authors are typically explaining complex problems, thus my request in respect of the Burgers model. Because I am a bit confused, sitting in between books during this review.

I have learnt about the model itself in the past being built on lambda as a parameter of the model. Lambda is often referred to as viscosity, but in fact it is viscous traction (as far as I know). In a way it is a viscosity but measured in one specific way during tensile loading. There is a relationship between viscosity (eta) and viscous traction (lambda), first studied by Couette.

I am at the moment sitting in front of 3 books dealing with rheology and few articles that present solutions, and unfortunately it seems that every author has their own take on the subject of eta and lambda for Burgers, and sometimes they are used as equal. Thus this creates a need for better referencing on hereby authors side to explain their chosen path or refer to the solution elsewhere. In my opinion, if your experiment is measuring viscous traction then the viscosity value will be slightly different when equation is solved. As far as I can see TCT is measuring the viscous traction, but is the bending beam measuring the same? I think this is more of a shear situation for BBC. I think Burgers solution for both of them could be different, and this could be the reason why TCT is on some level making sense, but BBC seems to be giving odd results from the calculations for the retarded element? Perhaps the ratio of measured modulus has to do with that as well, the different viscosity measurement?

Answer to comment 2: The authors agree with the reviewer regarding the Burgers model interpretation. As we stated in the paper, we thought that TCT gives more reliable values, especially for the retarded elements, due to simpler stress conditions in the test. Bending test with more complicated stress conditions gave values of E2 and h2 parameters that were, in our opinion, underestimated. It sometimes resulted in variable results in function of the temperature. In the case of TCT test results this situation is not visible. Determination of the properties was based on the methods developed by prof. Judycki. On the basis of TCT and BBCT results, rheological parameters describing elastic and viscous properties of the asphalt mixtures were determined from creep curves. Each of the creep curves was described using Equation 6 presented in the paper. Burgers parameters were treated as fitting parameters in the optimization function using the least square error methods. Fig. 6 - the creep curves and fitting of the creep using Burgers model shows the accuracy of the used optimization function. As for l in the Burgers equation, as stated in the paper, it is a function of retarded parameters. We do not treat it as equal to eta. We treat it as a different property, where l gives information regarding the stress relaxation.

3.     Language wise, the text between lines 230-253 has a surprising number of typos in comparison to the rest of the paper that seemed to be almost perfectly edited. I suggest revision of that chapter. And the correct word in English is "asymptote".

Answer to comment 3: All the typos have been corrected.

Reviewer 2 Report

The paper can be accepted for publication after minor changes. The topic of the article relates well to the themes of this Journal. The research methodology is robust, and the results and subsequent discussion seem to be significant. Nevertheless, authors should explain in more detail the innovative nature of this paper, taking into account the previous publication of several articles on closely related topics. Describe the real contributions of this paper for the knowledge about the low-temperature performance of asphalt mixtures (i.e., is it essential to make a comparative evaluation of TCT and BBC to assess thermal stresses in asphalt mixtures at low temperatures?)

The previous experience of the authors on the study of low-temperature behavior of asphalt mixtures, with several papers published on closely related topics (also cited in this paper), resulted in a well-organized document, with few features to be corrected. I didn’t find plagiarism or republication problems. Nevertheless, the close association of this paper with other published in previous years, some of them used as “feeding” material for this document, must be observed. Authors should analyze the possible overlap with those previous papers, and erase any potential repetitions.

The paper is unnecessarily long. Some parts of the introduction are not necessary because they are presenting aspects that are not relevant for this paper (e.g., the low-temperature performance of cement concrete pavements; grouped citation for a single idea). Also, some parts of the document can be shortened, simplified or probably erased, like the construction of master curves using Arrhenius and WLF equations.

The innovative nature and importance of the approach presented in this paper must be clearly stated, explaining its main benefits. After reading the article, I am not sure if it is better to use one or another test, how much it is essential to have a good relationship of each one of the tests with existing models. Some questions to be addressed are the following:

- Is the model more precise than experiments?

- Is the error of some experiments (40%) responsible for differences in the observed results?

- Can authors justify the differences in the performance at low temperatures to material changes occurring near the glass transition temperature of asphalt binders?

- Are those differences caused by slippery of asphalt specimens in the BBC test or by the differential distribution of stresses in TCT tests when specimen tops are not 100% parallel?

The organization of the article is right but can be improved. Mainly, the description of the materials should be developed and the paper should be shortened. The discussion of results is generally good, but further scientific additions to some results are welcome addressing the questions presented above. Written English quality is satisfying. The objectives are clear, and the paper answers those objectives, although their importance should be explained in more detail.

Some specific problems or questions to be answered are related to:

- The variation of test results was presented as varying between 1 and 40%. And in some cases, only two tests were performed in TCT. BBC used five specimens, but that is not a standard procedure for evaluation of the low-temperature performance of asphalt mixtures, namely the induced thermal stresses. Thus, please present more details on laboratory difficulties during specimen preparation and test performance, and relate those difficulties with the variation of up to 40% obtained in the results.

- The master curves presented in Figs. 12, 15 and 16 do not show the partial test results obtained at different temperatures. Figs. 13 and 14 give the partial results, but without a proper adjustment. Authors should evaluate the need for this part of the paper, and explain if the superposition was always like in Figs. 13 and 14. Also state if the changing phase of the asphalt materials near the glassing temperatures is responsible for some problems in the adjustment of the master curves, or if it is related to problems associated with test procedures.

Author Response

We would like to express our thanks to the reviewer for the effort put into evaluation of our paper. The remarks have been taken into consideration and they will certainly improve the paper.

Comments and Suggestions of Reviewer 2:

“The paper can be accepted for publication after minor changes. The topic of the article relates well to the themes of this Journal. The research methodology is robust, and the results and subsequent discussion seem to be significant. Nevertheless, authors should explain in more detail the innovative nature of this paper, taking into account the previous publication of several articles on closely related topics. Describe the real contributions of this paper for the knowledge about the low-temperature performance of asphalt mixtures (i.e., is it essential to make a comparative evaluation of TCT and BBC to assess thermal stresses in asphalt mixtures at low temperatures?)

The previous experience of the authors on the study of low-temperature behavior of asphalt mixtures, with several papers published on closely related topics (also cited in this paper), resulted in a well-organized document, with few features to be corrected. I didn’t find plagiarism or republication problems. Nevertheless, the close association of this paper with other published in previous years, some of them used as “feeding” material for this document, must be observed. Authors should analyze the possible overlap with those previous papers, and erase any potential repetitions.

The paper is unnecessarily long. Some parts of the introduction are not necessary because they are presenting aspects that are not relevant for this paper (e.g., the low-temperature performance of cement concrete pavements; grouped citation for a single idea). Also, some parts of the document can be shortened, simplified or probably erased, like the construction of master curves using Arrhenius and WLF equations.

The innovative nature and importance of the approach presented in this paper must be clearly stated, explaining its main benefits. After reading the article, I am not sure if it is better to use one or another test, how much it is essential to have a good relationship of each one of the tests with existing models. Some questions to be addressed are the following:

- Is the model more precise than experiments?

- Is the error of some experiments (40%) responsible for differences in the observed results?

- Can authors justify the differences in the performance at low temperatures to material changes occurring near the glass transition temperature of asphalt binders?

- Are those differences caused by slippery of asphalt specimens in the BBC test or by the differential distribution of stresses in TCT tests when specimen tops are not 100% parallel?

The organization of the article is right but can be improved. Mainly, the description of the materials should be developed and the paper should be shortened. The discussion of results is generally good, but further scientific additions to some results are welcome addressing the questions presented above. Written English quality is satisfying. The objectives are clear, and the paper answers those objectives, although their importance should be explained in more detail.

Some specific problems or questions to be answered are related to:

- The variation of test results was presented as varying between 1 and 40%. And in some cases, only two tests were performed in TCT. BBC used five specimens, but that is not a standard procedure for evaluation of the low-temperature performance of asphalt mixtures, namely the induced thermal stresses. Thus, please present more details on laboratory difficulties during specimen preparation and test performance, and relate those difficulties with the variation of up to 40% obtained in the results.

- The master curves presented in Figs. 12, 15 and 16 do not show the partial test results obtained at different temperatures. Figs. 13 and 14 give the partial results, but without a proper adjustment. Authors should evaluate the need for this part of the paper, and explain if the superposition was always like in Figs. 13 and 14. Also state if the changing phase of the asphalt materials near the glassing temperatures is responsible for some problems in the adjustment of the master curves, or if it is related to problems associated with test procedures.”

Answers to the comments and suggestions. The authors would like to thank the reviewer for the general comments. Regarding the detailed suggestions, our answers are as follows:

1.       The paper can be accepted for publication after minor changes. The topic of the article relates well to the themes of this Journal. The research methodology is robust, and the results and subsequent discussion seem to be significant. Nevertheless, authors should explain in more detail the innovative nature of this paper, taking into account the previous publication of several articles on closely related topics. Describe the real contributions of this paper for the knowledge about the low-temperature performance of asphalt mixtures (i.e., is it essential to make a comparative evaluation of TCT and BBC to assess thermal stresses in asphalt mixtures at low temperatures?)

Answer to comment 1: The explanation of the innovative nature of the paper has been added to the text:

The innovative nature of this paper is the application of tensile creep test procedure and its evaluation for the assessment of low temperature properties. The nature of thermal stresses that are built up in the asphalt pavement when the temperature decreases is tension rather than bending. Most of the research projects concerning thermal stresses calculation are based on bending tests. A more detailed study of tensile behavior and its comparison with previous test results based on three point bending test seems to be important.

2.   The previous experience of the authors on the study of low-temperature behavior of asphalt mixtures, with several papers published on closely related topics (also cited in this paper), resulted in a well-organized document, with few features to be corrected. I didn’t find plagiarism or republication problems. Nevertheless, the close association of this paper with other published in previous years, some of them used as “feeding” material for this document, must be observed. Authors should analyze the possible overlap with those previous papers, and erase any potential repetitions.

Answer to comment 2: We are grateful for the reviewer’s interest in our research. This paper was the next step in our research for better understanding of the low-temperature properties of asphalt mixtures. The possible overlap with previous papers has been analysed. In previous studies, for example in the paper [6] Pszczola, M.; Szydlowski C., Influence of Bitumen Type and Asphalt Mixture Composition on Low-Temperature Strength Properties According to Various test Methods, Materials – MDPI, (2018, 11, 2118, doi:10.3390/ma11112118), the authors tested the same asphalt mixtures as in the reviewed paper. So the properties of the mixtures are the same. The idea, objectives, results and discussion are, however, completely different. Also the previous paper in Materials: [39] Pszczola, M.; Jaczewski, M.; Rys, D., Jaskula, P.; Szydlowski, C., Evaluation of Asphalt Mixture Low-Temperature Performance in Bending Beam Creep Test, Materials – MDPI (2018, vol. 11, 100, https://doi.org/10.3390/ma11010100) addressed the subject of assessment of the bending beam properties of asphalt mixtures at low temperatures.

3.       The paper is unnecessarily long. Some parts of the introduction are not necessary because they are presenting aspects that are not relevant for this paper (e.g., the low-temperature performance of cement concrete pavements; grouped citation for a single idea). Also, some parts of the document can be shortened, simplified or probably erased, like the construction of master curves using Arrhenius and WLF equations.

Answer to comment 3: The authors agree with the reviewer regarding the need to shorten the paper. All the possible corrections have been introduced. Irrelevant sections have been removed.

4.       The innovative nature and importance of the approach presented in this paper must be clearly stated, explaining its main benefits. After reading the article, I am not sure if it is better to use one or another test, how much it is essential to have a good relationship of each one of the tests with existing models. Some questions to be addressed are the following:

- Is the model more precise than experiments?

- Is the error of some experiments (40%) responsible for differences in the observed results?

- Can authors justify the differences in the performance at low temperatures to material changes occurring near the glass transition temperature of asphalt binders?

- Are those differences caused by slippery of asphalt specimens in the BBC test or by the differential distribution of stresses in TCT tests when specimen tops are not 100% parallel?

Answer to comment 4: The authors agree with the reviewer. The innovative nature and importance of the approach presented in this paper have been explained in a passage added to the Introduction.

- The model is not as precise as the experiments. In the model only the mean values were used. We agree that for a comprehensive analysis also the variation in results should be analysed, as the thermal cracks appear in the weakest point of the pavement. Analysis presented in the article was used for evaluation of the TCT test for the purpose of cryogenic stress analysis using common models.

- The presented error (40%) appeared only in specific situations – tests at the highest temperatures, where the absolute values of stiffness are small. In this case even small absolute differences, which won’t influence the calculations, result in quite high relative differences. For the most cases, the variation is around 10-15%, which was illustrated by mean value for each test.

- The differences in the performance at low temperatures to material changes occurring near the glass transition temperature of asphalt binders were not the subject of the presented research.

- Those differences could have resulted from the different distribution of stress in TCT tests. The influence of specimen imperfections, however, has not yet been analysed. In our opinion it can be a good idea to investigate that issue in the future.

5.    The organization of the article is right but can be improved. Mainly, the description of the materials should be developed and the paper should be shortened. The discussion of results is generally good, but further scientific additions to some results are welcome addressing the questions presented above. Written English quality is satisfying. The objectives are clear, and the paper answers those objectives, although their importance should be explained in more detail.

Answer to comment 5: The authors would like to thank the reviewer for the comment. All the possible corrections have been introduced. The description of the materials is very limited and, in the authors’ opinion, could not be shortened.

6.         Some specific problems or questions to be answered are related to:

- The variation of test results was presented as varying between 1 and 40%. And in some cases, only two tests were performed in TCT. BBC used five specimens, but that is not a standard procedure for evaluation of the low-temperature performance of asphalt mixtures, namely the induced thermal stresses. Thus, please present more details on laboratory difficulties during specimen preparation and test performance, and relate those difficulties with the variation of up to 40% obtained in the results.

- The master curves presented in Figs. 12, 15 and 16 do not show the partial test results obtained at different temperatures. Figs. 13 and 14 give the partial results, but without a proper adjustment. Authors should evaluate the need for this part of the paper, and explain if the superposition was always like in Figs. 13 and 14. Also state if the changing phase of the asphalt materials near the glassing temperatures is responsible for some problems in the adjustment of the master curves, or if it is related to problems associated with test procedures.”

Answer to comment 6:

-        Yes, the variation of test results that were presented varied between 1 and 40%. The problem with TCT test procedure is that it needs more time to test a single specimen. According to the TCT procedure applied, it was 8 hours of loading and 2 hours of unloading. So only 1 specimen could be tested each day. That was the reason why only 2 test specimens were tested at each temperature for a given asphalt mixture. We believe that it is a good idea to limit the loading time in future tests, but additional research is needed for TCT test procedure. The limitations of the research conducted were presented in Summary and Conclusions (point 8).

-        The problem presented in figures from 12 to 16 was explained in detail in the paper: “Jaczewski M., Judycki J., Jaskula P., Asphalt concrete subjected to long-time loading at low temperatures – deviations from the time-temperature superposition principle, Construction and Building Materials, vol. 202, 2019, p. 426-439, https://doi.org/10.1016/j.conbuildmat.2019.01.049“. As the reviewer stated before, we took care not to overlap the material presented in both papers. The adjustments presented in the Figures 13 and 14 are proper, taking into consideration the assumptions presented in the recalled article. For long times of loading (>1000 seconds) the curves tend to deviate from uniform lines, therefore it seems that they are not properly shifted. Typical low temperature creep tests end before 1000 seconds, so the deviations are not visible. The master curves presented in the Figures 15 and 16 show only the “main” master curve line. If it is required we will add additional “branches” of the master curve.

-        Figure 12 does not show master curves. It presents only selected stiffness curves for the temperatures of -10 and -20°C without any shifting.

-        As stated, the problem is not related to the test procedure. The problem might be related to the glass transition, where the material is behaving more like elastic material than viscoelastic material. But the viscous flow is still visible even at the temperatures of -20 or -30°C.

-        The authors agree with the reviewer. Some figures have been removed from the paper and appropriate explanation has been added to the paper.
